# Autocrine regulation of stomatal differentiation potential by EPF1 and ERECTA-LIKE1 ligand-receptor signaling

Xingyun Qi[1,2], Soon-Ki Han[1,2], Jonathan H Dang[1,2], Jacqueline M Garrick[1,2], Masaki Ito[3], Alex K Hofstetter[1,2], Keiko U Torii[1,2,3]*

[1]Howard Hughes Medical Institute, University of Washington, Seattle, United States; [2]Department of Biology, University of Washington, Seattle, United States; [3]Graduate School of Bioagricultural Sciences/Institute of Transformative Biomolecules (WPI-ITbM), Nagoya University, Nagoya, Japan

**Abstract** Development of stomata, valves on the plant epidermis for optimal gas exchange and water control, is fine-tuned by multiple signaling peptides with unique, overlapping, or antagonistic activities. EPIDERMAL PATTERNING FACTOR1 (EPF1) is a founding member of the secreted peptide ligands enforcing stomatal patterning. Yet, its exact role remains unclear. Here, we report that EPF1 and its primary receptor ERECTA-LIKE1 (ERL1) target MUTE, a transcription factor specifying the proliferation-to-differentiation switch within the stomatal cell lineages. In turn, MUTE directly induces *ERL1*. The absolute co-expression of ERL1 and MUTE, with the co-presence of EPF1, triggers autocrine inhibition of stomatal fate. During normal stomatal development, this autocrine inhibition prevents extra symmetric divisions of stomatal precursors likely owing to excessive MUTE activity. Our study reveals the unexpected role of self-inhibition as a mechanism for ensuring proper stomatal development and suggests an intricate signal buffering mechanism underlying plant tissue patterning.

*For correspondence: ktorii@u.washington.edu

**Competing interests:** The authors declare that no competing interests exist.

## Introduction

Developmental patterning of multicellular organisms relies on positional cues as well as local cell-cell interactions. Development of stomata, turgor-driven cellular valves on the plant epidermis for optimal gas exchange while minimizing water loss, is coordinated by the intricate cell-cell signaling that impinges on the activity of cell-fate determinants (*Pillitteri and Dong, 2013*; *Han and Torii, 2016*). Stomatal development in *Arabidopsis* occurs through stereotypical cell-state transition events: a stomatal initial cell on the protoderm called a meristemoid mother cell (MMC) undergoes an asymmetric entry division to give rise to a meristemoid, a transient amplifying cell of the stomatal lineage. The meristemoid reiterates asymmetric divisions and renews itself, while creating surrounding stomatal lineage ground cells (SLGCs). Eventually, the meristemoid loses its stem-cell-like property and differentiates into a guard mother cell (GMC). The GMC executes a single round of symmetric division to form paired guard cells (GCs) surrounding a pore, thereby completing stomatal differentiation (*Lau and Bergmann, 2012*; *Pillitteri and Torii, 2012*; *Han and Torii, 2016*). These cell state transitions are directed by the sequential actions of three basic-helix-loop-helix (bHLH) proteins, SPEECHLESS (SPCH), MUTE, and FAMA, which determine the initiation and proliferation, meristemoid-to-GMC transition, and GMC-to-GC differentiation, respectively. These three bHLH proteins form heterodimers with the broadly expressed sister bHLH proteins SCREAM (SCRM: also known as ICE1) and SCRM2, both of which are absolutely required for the activities of SPCH-MUTE-FAMA (*Lau and Bergmann, 2012*; *Pillitteri and Dong, 2013*; *Han and Torii, 2016*).

The spatial control of stomatal differentiation processes is achieved by a family of secreted cyste-ine-rich peptides called EPIDERMAL PATTERNING FACTORS (EPFs) that are perceived by cell-sur-face receptors with extracellular leucine-rich repeat (LRR) domain: three ERECTA-family receptor kinases and their signal modulator LRR receptor-like protein TOO MANY MOUTHS (TMM) (*Nadeau and Sack, 2002*; *Shpak et al., 2005*; *Hara et al., 2007*, *2009*; *Hunt and Gray, 2009*; *Torii, 2012*). These modules are conserved down to basal land plants (*Caine et al., 2016*). During the initiation of stomatal cell lineages, EPF2 peptide perception by ERECTA activates downstream intracellular signals mediated by the mitogen-activated protein kinase (MAP kinase) cascade, which inhibits the entry divisions of MMC by down-regulating SPCH protein (*Lampard et al., 2008*; *Lee et al., 2012*, *2015*). As such, application of bioactive EPF2 peptide confers an epidermis solely composed of pavement cells, phenotypically identical to loss-of-function *spch* mutant (*Lee et al., 2012*). Recent studies established that *EPF2* is a direct transcriptional target of SPCH and SCRM, therefore EPF2, ERECTA•TMM, and SPCH•SCRM modules constitute a negative feedback loop gen-erating evenly distributed spatial patterns of stomatal initial cells (*Lau et al., 2014*; *Horst et al., 2015*). Furthermore, the decision of whether or not to enter the stomatal cell-lineage is fine-tuned by the competitive binding of EPF2 and its antagonistic peptide, EPF-LIKE9 (EPFL9), also known as Stomagen, to ERECTA (*Kondo et al., 2010*; *Sugano et al., 2010*; *Lee et al., 2015*). Unlike EPF2, the direct binding of Stomagen to ERECTA leads to the inhibition of receptor activation and signal transduction, therefore resulting in excessive stomatal clusters like *erecta*-family mutants (*Lee et al., 2015*).

The founding member of EPF/EPFL-family peptides, EPF1, enforces stomatal spacing and its loss-of-function mutation confers stomatal pairing and clustering (*Hara et al., 2007*). Both genetic and biochemical studies have placed ERECTA-LIKE1 (ERL1), a sister receptor of ERECTA, as well as TMM as a receptor module acting downstream of EPF1 (*Hara et al., 2007*; *Lee et al., 2012*). However, unlike EPF2, the exact action and function of EPF1 remain unclear. Early models predicted that the signal emanating from a meristemoid is perceived by the receptors in the neighboring cells to regu-late stomatal patterning (*Nadeau and Sack, 2003*; *Serna, 2004*). However, both *EPF1* and *ERL1* promoter activities are the highest in the stomatal precursor cells, most notably in the late meriste-moids and GMCs (*Shpak et al., 2005*; *Hara et al., 2007*). The inconsistency of their expression pat-terns and predicted functions adds to the conundrum as to how this ligand-receptor system operates. It is also unknown whether EPF1 signaling acts on any of the stomatal bHLH transcription factors. To address these questions, we investigated the in vivo dynamics and regulation of ERL1, the primary receptor of EPF1. We found that ERL1 and MUTE constitute a negative feedback loop, which, with co-expression of EPF1, elicits an autocrine inhibition of stomatal differentiation potential. Our study reveals the unexpected role of self-inhibition as a mechanism for ensuring timely transition of stomatal differentiation and proper stomatal patterning. Given that EPF1 peptide sources are shared by both autocrine and paracrine signaling, the intricate buffering mechanism may underpin proper stomatal patterning.

## Results

### ERL1 exhibits dynamic accumulation during asymmetric divisions of stomatal precursors and meristemoid-to-GMC transition

To clarify the role of EPF1-ERL1 signaling during stomatal development, we first examined the in vivo dynamics of ERL1-YFP fusion protein driven by its promoter expressed in *erl1-2* null mutant background (*ERL1pro::ERL1-YFP in erl1*). This construct rescues the stomatal cluster phenotype in *er erl1 erl2*, indicating it is functional (*Figure 1—figure supplement 1*). Consistently, the expression level of the *ERL1-YFP* transgene is comparable to that of the endogenous *ERL1* (*Figure 1—figure supplement 1D*). In early stomatal-lineage cells, including rectangular MMCs and early meriste-moids, ERL1-YFP is localized sharply at the plasma membrane (*Figure 1A–C,K,L*; *Video 1*). During the asymmetric divisions of stomatal-lineage cells, ERL1-YFP exhibited striking accumulation at the newly-formed division site that separates the meristemoid and stomatal lineage ground cell (SLGC) (*Figure 1D,F,M*; *Video 1*).

Following the division, ERL1-YFP signals were detected in the plasma membrane of both the mer-istemoid and neighboring stomatal lineage ground cell (SLGC) (*Figure 1E,G,N*; *Video 1*). During the

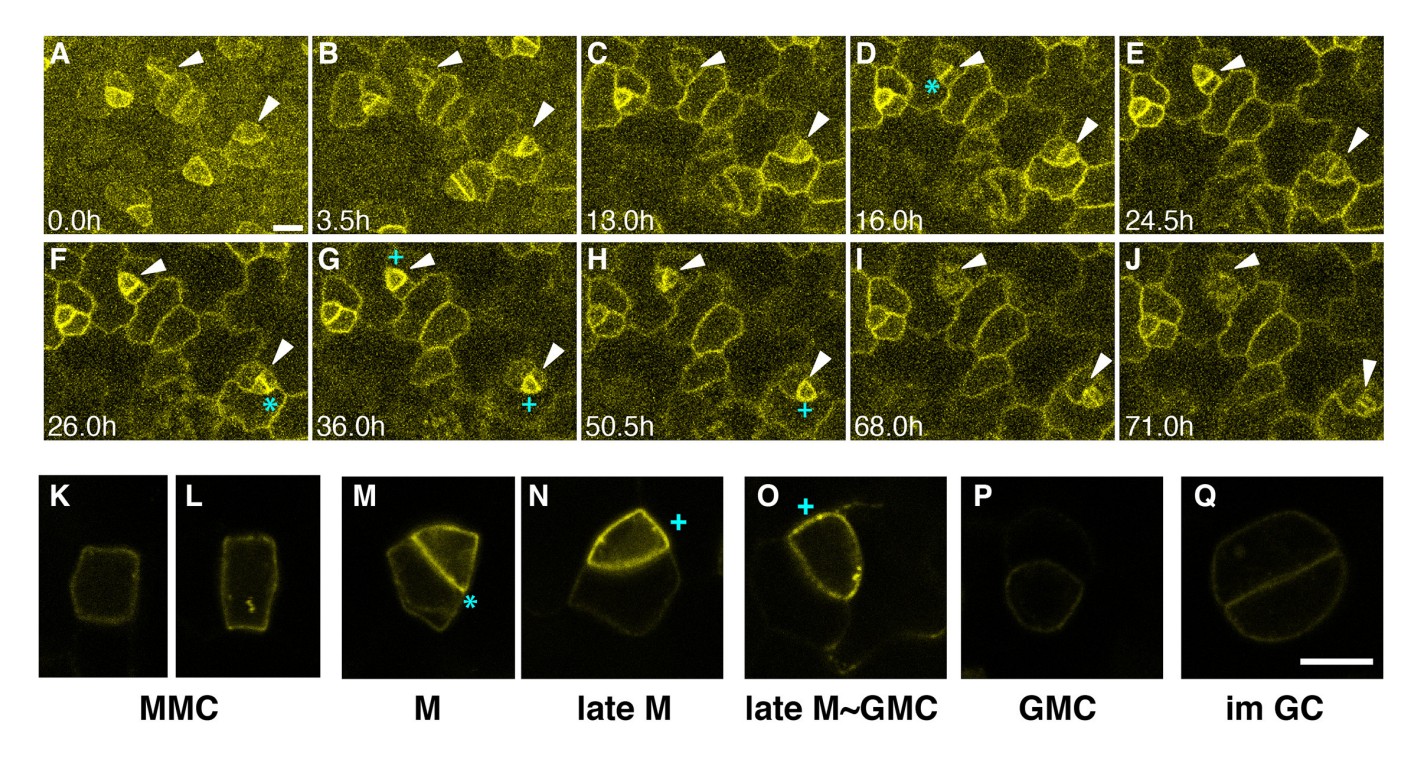

**Figure 1.** Expression patterns of ERL1 during stomatal development. Expression pattern of *ERL1pro::ERL1-YFP*. (A–J) Time-lapse live imaging of developing abaxial cotyledon epidermis of the 1-day-old T4 seedling of *ERL1pro::ERL1-YFP erl1-2*. Time points after image collection are indicated in hours:(A) 0.0 hr; (B) 3.5 hr; (C) 13.0 hr; (D) 16.0 hr; (E) 24.5 hr; (F) 26.0 hr; (G) 36.0 hr; (H) 50.5 hr; (I) 68.0 hr; and (J) 71.0 hr. Arrowheads point to two representative cells. Images for (A–J) are taken at the same magnification. Scale bar, 10 µm. (K–L) High resolution live images of stomatal precursors expressing ERL1-YFP. (K, L) meristemoid mother cells (MMC); (M) meristemoid (M); (N) late meristemoid (late M); (O) late meristemoid to guard mother cell transition (late M~GMC); (P) GMC; (Q) immature guard cells (im GC). Images for (K–L) are taken at the same magnification. Scale bar, 7.5 µm. ERL1-YFP signals are detected at the plasma membrane from early meristemoids (A, K, L; arrowheads) and accumulate high at the asymmetric division site (e.g. D, F, M; cyan asterisks). ERL1-YFP signals boost during the late meristemoid-to-GMC transition (G, H, N, O; cyan plus), and diminishes after GMC symmetric division (I, J, P, Q; arrowheads). See accompanying *Video 1*. Experiments were repeated three times. Total seedlings analyzed; n = 9.

The following figure supplement is available for figure 1:

**Figure supplement 1.** *ERL1pro::ERL1-YFP* is functional.

late meristemoid-to-GMC transition, ERL1-YFP showed transient yet high accumulation in the meristemoids while diminishing from the stomatal lineage ground cells (SLGCs), marking the late meristemoids the brightest (*Figure 1G,H,N,O*; *Video 1*). Expression of ERL1 disappears as the GMC divides symmetrically and differentiates into guard cells (*Figure 1J,P,Q*; *Video 1*).

### *ERL1* is a direct MUTE target

It has been reported that *ERL1* is a high-confidence SPCH target gene (*Lau et al., 2014*). While this is consistent with the early stomatal-lineage expression of ERL1, it does not explain the rapid boost in accumulation of ERL1 during the meristemoid-to-GMC transition (*Figure 1*). The transition from meristemoids to GMCs is specified by SPCH's sister bHLH protein, MUTE (*Pillitteri et al., 2007*). We thus hypothesized that MUTE is driving the rapid increase of ERL1 in the late meristemoids. To address this, we first examined the MUTE and ERL1 proteins in the stomatal lineage cells using *Arabidopsis* lines co-expressing *MUTE-tagRFP* and *ERL1-YFP* driven by their own respective endogenous promoters (*Figure 2A*). As expected, early stomatal-lineage cells only express ERL1-YFP. However, all late meristemoids co-expressed MUTE-tagRFP in the nuclei and strong ERL1-YFP at the cell periphery (n = 270; 12 seedlings analyzed) (*Figure 2A,B*). MUTE-tagRFP was driven by the

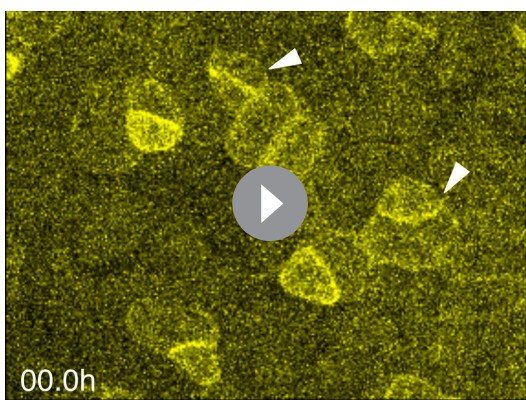

**Video 1.** Time-lapse movie of ERL1-YFP dynamics during stomatal development. Projection of z-stack images of *ERL1pro::ERL1-YFP* from the abaxial cotyledon epidermis of the 1-day-old T4 seedling were shown in 30 min time interval. Time points after image collection are indicated in hours. Arrowheads point to two representative cells.

identical promoter (−1989 bp) used to express MUTE-GFP (*Pillitteri et al., 2007*). Both *MUTE-pro::MUTE-GFP* and *MUTEpro::MUTE-tagRFP* exhibited transient signals during late meristemoid-to-GMC transition (*Figure 2—figure supplement 1A*), and their co-introduction into Arabidopsis seedlings further confirmed their co-expression (*Figure 2—figure supplement 1B*). Thus, the MUTE-tagRFP line accurately reflects the previously-documented MUTE dynamics.

Next, we used the estrogen receptor-based inducible overexpression system (XVE system) (*Zuo et al., 2000*) to address if *ERL1* is upregulated by MUTE. The estradiol-inducible *MUTE* overexpression (*iMUTE*) line upregulates *MUTE* transcripts within two hours and remained high throughout the analyzed time course, with the maximum average fold increase of 1536 ± 79 after 8 hr (*Figure 2C*). *iMUTE* triggered rapid upregulation in *ERL1* transcripts also within two hours, indicating that *ERL1* is an early MUTE-induced gene (*Figure 2D*; *Figure 2—figure supplement 2E*). To unequivocally address whether *ERL1* is a direct MUTE target, a chromatin immunoprecipitation (ChIP) assay was performed. For this purpose, we took advantage of the *Arabidopsis scrm-D* enabled line that vastly increases the stomatal-lineage cell population thereby enhancing the signals for ChIP assays (*Horst et al., 2015*). Our previous study has shown that the *scrm-D*-enabled transcriptomic profiling of meristemoids accurately reflects the meristemoid-specific gene expression during normal epidermal development (*Pillitteri et al., 2011*), further supporting the use of this genetic background. We introduced functional MUTE-GFP protein driven by its own promoter (*MUTEpro::MUTE-GFP*) (*Pillitteri et al., 2007*), into *scrm-D* mutant background. A robust binding of MUTE was detected within the *ERL1* promoter at the region overlapping with the known SPCH binding site (*Lau et al., 2014*), which contains the predicted bHLH-binding E-box (*Figure 2H,I*; *Figure 2—figure supplement 2A,B,D*). Furthermore, the binding of MUTE-GFP at the *ERL1* promoter region was also detected in the complemented genetic background, *MUTEpro::MUTE-GFP* expressed in *mute*, which exhibits wild-type phenotype (*Pillitteri et al., 2007*). As predicted, the fold-enrichment of signals were reduced in the absence of *scrm-D* mutation (*Figure 2—figure supplement 2C*). The data demonstrate the binding of MUTE-GFP during normal epidermal development. Collectively, the results indicate that MUTE induces *ERL1* expression by directly associating with the *ERL1* promoter region.

## Expression of EPF1, the primary ligand for ERL1, overlaps with MUTE during the developmental window of stomatal precursor cells

It has been reported that ERL1 primarily perceives EPF1 peptides (*Hara et al., 2007*; *Lee et al., 2012*). To clarify the spatiotemporal expression of EPF1 and MUTE, we next analyzed dual reporter seedlings expressing *MUTEpro::MUTE-tagRFP* and *EPF1pro::erGFP*. A population of meristemoids only exhibited nuclear RFP signals, suggesting that the onset of *MUTE* expression precedes that of EPF1 (14/180 cells expressed RFP only; from five seedlings) (*Figure 2F,G*). In the late meristemoids and GMCs, strong RFP and GFP signals co-exist, indicating that MUTE and EPF1 are co-expressed (*Figure 2F,G*). The *EPF1* promoter activity remained very high in immature GCs, where MUTE-tagRFP signal was no longer detected (*Figure 2F,G*). We next analyzed whether *EPF1* is also upregulated by MUTE. Unlike *ERL1*, however, *iMUTE* induction did not trigger any substantial changes in *EPF1* expression (*Figure 2E*; *Figure 2—figure supplement 2E*), suggesting that *EPF1* is not likely a MUTE target.

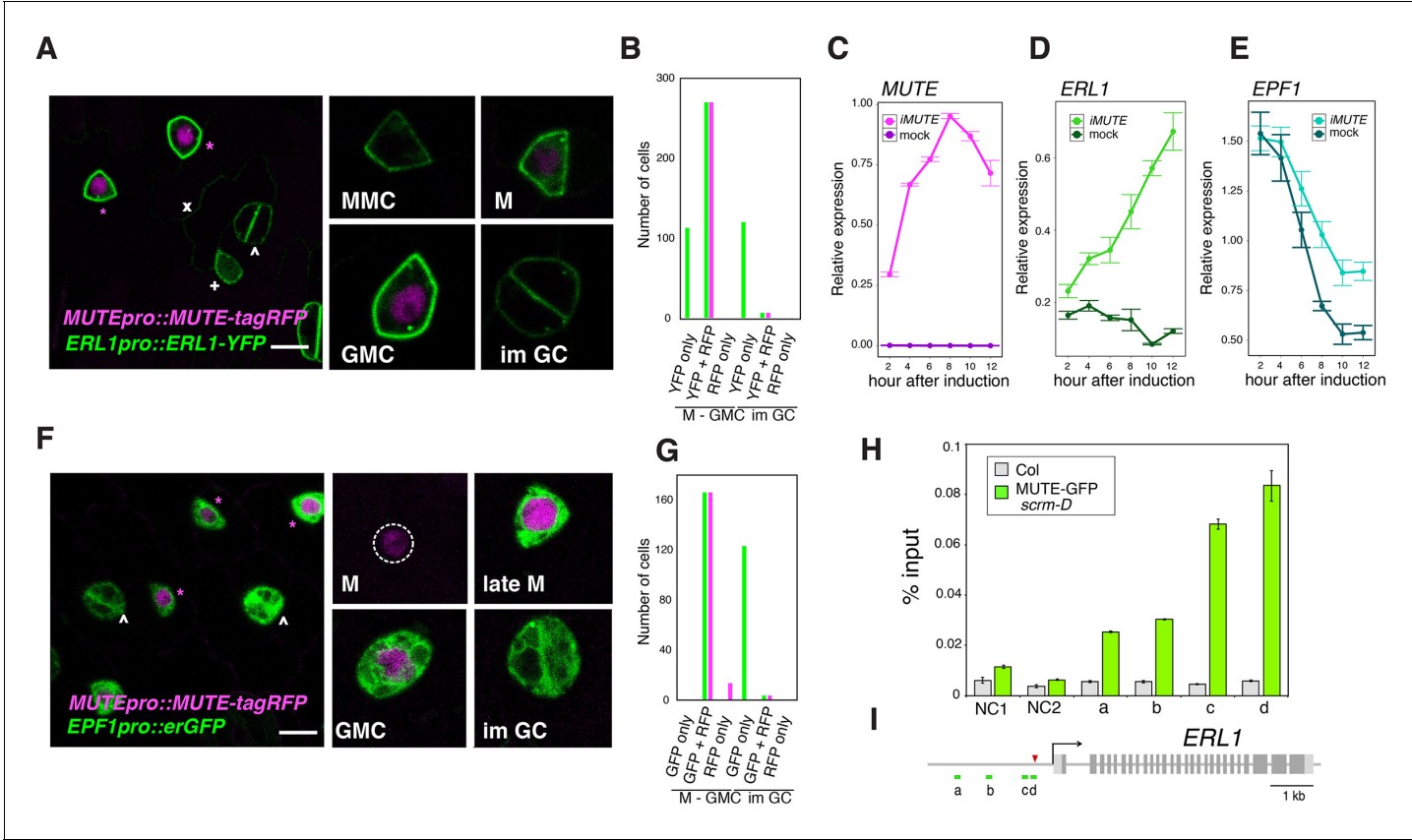

**Figure 2.** Overlapping expression of EPF1, ERL1, MUTE and direct upregulation of ERL1 by MUTE. (**A**) Expression patterns of *ERL1pro::ERL1-YFP* and *MUTEpro::MUTE-tagRFP* from the abaxial true leaf epidermis of 7-day-old F1 seedlings. Co-expression of MUTE-tagRFP in nucleus and ERL1-YFP at the plasma membrane are detected in late meristemoids and GMCs (top; magenta asterisks) while only ERL1-YFP signals are visible in MMC, early meristemoid (+) and immature GCs (^). Faint YFP signals are detected in the stomatal lineage ground cell (SLGC) differentiating into a pavement cell (x). (Insets) representative images of stomatal precursor cells. Scale bars, 7.5 μm. (**B**) Quantitative analysis of the number of stomatal precursor cells (meristemoids-to-GMC, and immature GCs) expressing *ERL1pro::ERL1-YFP* (green) and/or *MUTEpro::MUTE-tagRFP* (magenta). Total of 510 cells were analyzed from 12 seedlings. (**C**) Quantitative RT-PCR analysis of the total *MUTE* transcripts from 4-day-old seedlings that were either mock- or estradiol-treated for induced *MUTE* overexpression (*iMUTE*) at the time indicated. Relative expression represents qRT-PCR expression normalized over that of *ACTIN2*. Values are Mean ± standard deviation of three technical replicates from one representative experiment. (**D**) Quantitative RT-PCR analysis of the endogenous *ERL1* transcripts from 4-day-old seedlings that were either mock- or *iMUTE* at the time indicated. Relative expression represents qRT-PCR expression normalized over that of *ACTIN2*. Values are Mean ± standard deviation of three technical replicates from one representative experiment. See *Figure 2—figure supplement 2* for the two additional biological replicates. (**E**) Quantitative RT-PCR analysis of the endogenous *EPF1* transcripts from 4-day-old seedlings that were either mock- or *iMUTE* at the time indicated. Relative expression represents qRT-PCR expression normalized over that of *ACTIN2*. Values are Mean ± standard deviation of three technical replicates from one representative experiment. See *Figure 2—figure supplement 2* for the two additional biological replicates. (**F**) Expression patterns of *EPF1pro::erGFP* and *MUTEpro::MUTE-tagRFP* in the abaxial true leaf epidermis from 7-day-old F1 seedlings. MUTE-tagRFP expression precedes that of EPF1. Strong co-expression of MUTE-tagRFP and EPF1 can be detected in the late meristemoids and GMCs (top; magenta asterisks), while only EPF1 signals are visible in immature GCs (^). (Insets) representative images of stomatal precursor cells. Scale bars, 7.5 μm. (**G**) Quantitative analysis of the number of stomatal precursor cells (meristemoids-to-GMC, and immature GCs) expressing *EPF1pro::erGFP* (green) and/or *MUTEpro::MUTE-tagRFP* (magenta). Total of 307 cells were analyzed from five seedlings. (**H**) Quantitative PCR on *ERL1* promoter region after chromatin Immunoprecipitation against anti-GFP antibody in Col and transgenic plants expressing MUTE-GFP in *scrm-D*. Values are Mean ± S.E.M. of percent-input DNA of three technical replicates from one representative experiment. See *Figure 2—figure supplement 1* for the other two biological replicates. NC, Negative Control; NC1, 5′ intergenic region of *ACTIN2*; NC2, promoter region of *AGAMOUS*; a, b, c and d, the *ERL1* loci indicated below in (**D**). For raw data, see *Supplementary file 3*. (**I**) Diagram of the *ERL1* loci tested. Green lines (a,b,c and d) below the *ERL1* schematic, regions amplified by qPCR; Red arrowhead, SPCH binding peak; gray line, intergenic region or intron; light gray box, 5′ or 3′ untranslated region; dark gray box, exon; black arrow, transcription start site. See *Figure 2—figure supplement 2* for the two additional biological replicates.

The following figure supplements are available for figure 2:

**Figure supplement 1.** Expression patterns of *MUTEpro::MUTE-GFP* and *MUTEpro::MUTE-tagRFP*.

*Figure 2 continued on next page*

*Figure 2 continued*

**Figure supplement 2.** Additional two biological replicates of ChIP assays and qRT-PCR expression analysis of *ERL1* and *EPF1* upon induced *MUTE* overexpression.

## EPF1-ERL1 signaling pathway inhibits MUTE

Our finding that MUTE directly induces *ERL1* expression during meristemoid-to-GMC transition (*Figure 2*) implies a previously-undescribed role of the ERL1 receptor kinase in differentiating GMCs. *EPF1* overexpression or exogenous EPF1 peptide application confers arrested meristemoids, a phenotype resembling *mute*, in ERL1-dependent manner (*Lee et al., 2012*). To understand the ultimate target of this EPF1-ERL1 signaling, we examined the effects of excessive EPF1 on transcription factors driving GMC differentiation, MUTE and SCRM. For this purpose, we first made use of our estradiol-induced *EPF1* overexpression (*iEPF1*) line that rapidly upregulates *EPF1* transcript amounts to the maximum fold increase of $407 \pm 46$ at 5 hr after estradiol treatment (*Figure 3—figure supplement 1C*). As reported (*Hara et al., 2007*; *Lee et al., 2012*), *iEPF1* inhibits stomatal differentiation (*Figure 3—figure supplement 1A,B*). *iEPF1* conferred a loss of signals from both *MUTE* transcriptional (*MUTEpro::nucYFP*) and translational (*MUTEpro::MUTE-GFP*) reporters 4 days after induction when arrested meristemoid phenotypes are evident (*Figure 3A*). By contrast, signals for *SCRM* transcriptional (*SCRMpro::nucGFP*) and translational (*SCRMpro::GFP-SCRM*) reporters remained in arrested meristemoids even after 4 days (*Figure 3B*). The effects of EPF1 peptide application mirrored *iEPF1* (*Figure 3—figure supplement 2*). These results suggest that the EPF1 signal targets MUTE but not SCRM.

To understand the kinetics of MUTE disappearance, we next performed quantitative analysis of MUTE-GFP levels upon *iEPF1* induction hourly (*Figure 3C,D*). The MUTE-GFP signals started to diminish at 15 hr after the *iEPF1* induction and became barely detectable by 20 hr after the induction. By 24 hr, MUTE-GFP signals were undetectable (*Figure 3C,D*). In the mock-treated control, robust MUTE-GFP signals were detected throughout the 24 hr time course (*Figure 3C,D*). The timeframe is consistent with *iEPF1* transcript accumulation (*Figure 3—figure supplement 1C*). Within the same timeframe, transcriptional and translational reporters of SCRM were not influenced by *iEPF1* (*Figure 3E,F*). Gradual declines in GFP-SCRM signals were detected over time for both mock and *iEPF1*, which may be reflecting the developmental stage of the seedlings (*Figure 3E,F*).

To address the specificity of EPF1 signaling pathway, we further tested the effects of *iEPF1* on *SPCH* promoter and SPCH-GFP protein accumulation. Strong GFP signals of both *SPCHpro::nucGFP* and *SPCHpro::SPCH-GFP* were detected even three days after *iEPF1* overexpression (*Figure 4A*). By five days after *iEPF1* overexpression, SPCH-GFP protein was no longer detected (*Figure 4B*). This likely reflects the developmental progression of all remaining stomatal-lineage cells beyond the SPCH-regulated steps. Further quantitative analysis of SPCH transcriptional and translational reporters showed that their signals are not changed by *iEPF1* (*Figure 4C,D*). A steep decline in *SPCHpro::nucGFP* signals were detected within the first five hours in both mock and estradiol treatment for *iEPF1* (*Figure 4C*). This may imply that *SPCH* promoter activity is sensitive to the liquid treatment. The overall decline of signals may also reflect the developmental stage of the seedlings (*Figure 4C, D*). On the basis of these findings, we conclude that MUTE, but not SPCH, is the specific target of EPF1-mediated signaling.

## ERL1 mediates an autocrine inhibitory signaling of meristemoid-to-GMC differentiation

Our results indicate that ERL1 mediates extrinsic EPF1 peptide signaling, which in turn inhibits MUTE. Paradoxically, however, expression of the endogenous EPF1 coincides with MUTE (*Figure 2*). We thus postulated that expressing ERL1 only in the MUTE-expressing cells may lead to a loss of stomatal fate, in other words, having a secreted peptide, its receptor and target transcription factor only in the same cells will auto-inhibit stomatal differentiation. To test this hypothesis, we expressed the functional *ERL1-YFP* under the control of the *MUTE* promoter. *MUTEpro::ERL1-YFP* was introduced into *er erl1 erl2* triple null mutants in order to ensure that ERL1 will be the only existing

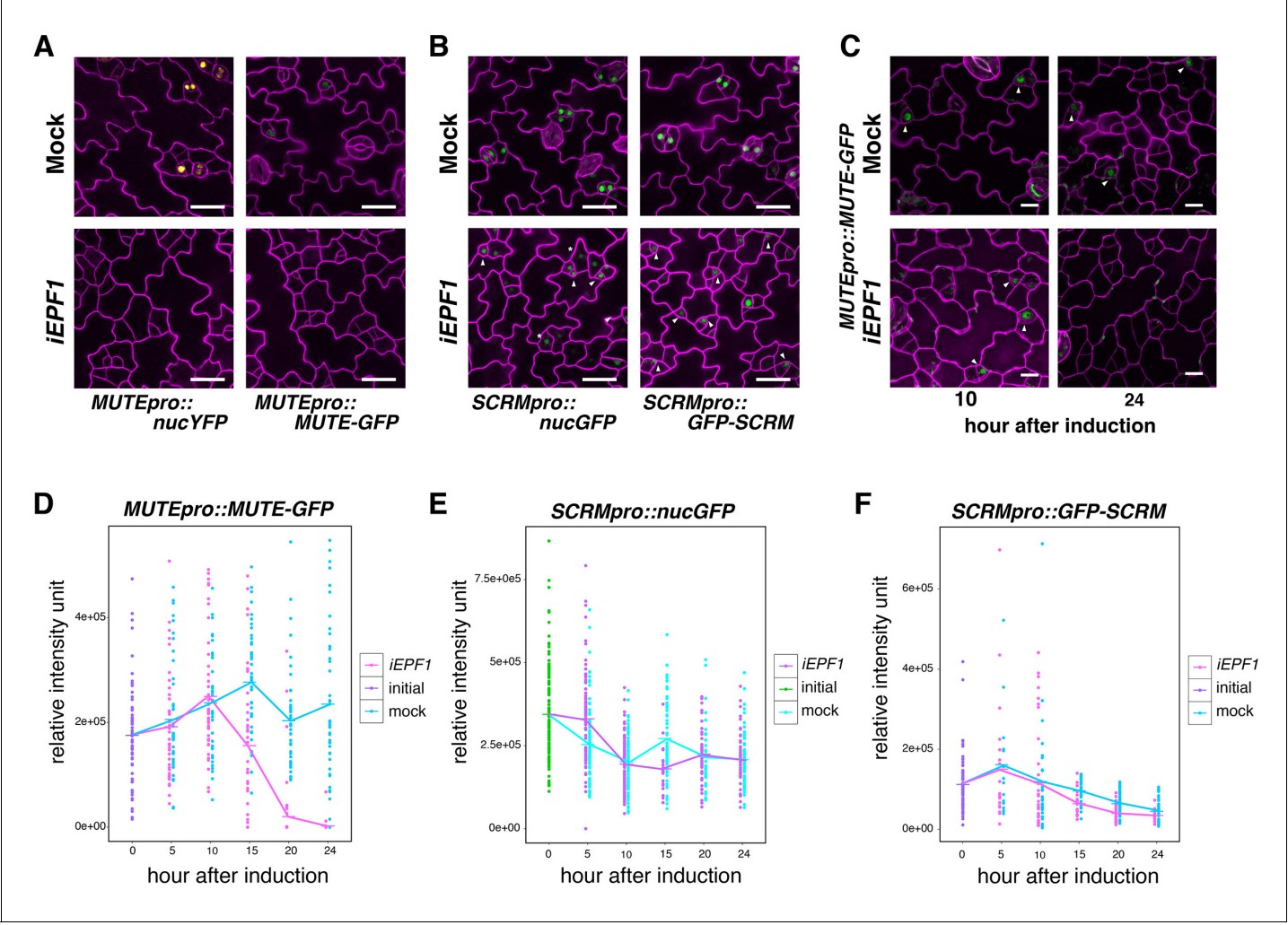

**Figure 3.** EPF1 signaling downregulates both *MUTE* promoter activity and MUTE protein accumulation. (A) Representative confocal images of abaxial cotyledon epidermis from 4-day-old estradiol-inducible EPF1 seedlings carrying *MUTEpro::nucYFP* and *MUTEpro::MUTE-GFP* four days after mock-treated or estradiol-treated for induced *EPF1* overexpression (*iEPF1*). F1 seedlings were used for the analysis. Scale bars, 25 μm. Experiments were repeated three times. Three seedlings were analyzed each time. (B) Representative confocal images of abaxial cotyledon epidermis from 4-day-old estradiol-inducible EPF1 seedlings carrying *SCRMpro::nucGFP* and *SCRMpro::GFP-SCRM* four days after mock-treated or estradiol-treated for induced *EPF1* overexpression (*iEPF1*). F1 seedlings were used for the analysis. Scale bars, 25 μm. Experiments were repeated three times. Three seedlings were analyzed each time. (C) Representative Z-stack confocal image projection showing the MUTE-GFP levels after iEPF1 induction used for the quantitative analysis. Meristemoids accumulating MUTE-GFP are indicated by arrowheads. F1 seedlings were used for the analysis. Scale bars, 10 μm. Experiments were repeated three times. Three seedlings were analyzed each time. (D) Quantitative analysis of MUTE-GFP levels after *iEPF1* induction at the time indicated on 3-day-old seedlings. Each dot represents the total intensity value of GFP signals from each nucleus expressing MUTE-GFP. To fully cover the entire nuclei expressing MUTE-GFP, serial Z-stack projection images were used for quantitative analysis (see Materials and methods). Mean value at each time point is connected by the line to visualize the average intensity change over time. Purple, initial values at time point 0; Cyan, mock treatment; Magenta, *iEPF1* induction. Experiments were repeated three times; n = 6 for time point 0; n = 3 per subsequent time point. Total of 495 nuclei were analyzed. (E) Quantitative analysis of SCRMpro::nucGFP levels after *iEPF1* induction at the time indicated on 3-day-old seedlings. Each dot represents the total intensity value of GFP signals from each nucleus expressing SCRMpro::nucGFP. To fully cover the entire nuclei expressing GFP, serial Z-stack projection images were used for quantitative analysis (see Materials and methods). Mean value at each time point is connected by the line to visualize the average intensity change over time. Green, initial values at time point 0; Cyan, mock treatment; Purple, *iEPF1* induction. Experiments were repeated three times; n = 6 for time point 0; n = 3 per subsequent time point. Total of 817 nuclei were analyzed. (F) Quantitative analysis of GFP-SCRM levels after *iEPF1* induction at the time indicated on 3-day-old seedlings. Each dot represents the total intensity value of GFP signals from each nucleus expressing SCRMpro:: GFP-SCRM. To fully cover the entire nuclei expressing GFP-SCRM, serial Z-stack projection images were used for quantitative analysis (see Materials and methods). Mean value at each time point is connected by the line to visualize the average intensity change over time. Purple, initial values at time point 0; Cyan, mock treatment; Magenta, *iEPF1* induction. Experiments were repeated three times; n = 6 for time point 0; n = 3 per subsequent time point. Total of 368 nuclei were analyzed.

*Figure 3 continued on next page*

*Figure 3 continued*

The following figure supplements are available for figure 3:

**Figure supplement 1.** Quantitative and phenotypic analyses of induced overexpression of *EPF1*.

**Figure supplement 2.** Predicted mature EPF1 peptide application reduces *MUTE* promoter activity and MUTE protein accumulation.

receptor pool that will perceive and transduce the signal (*Figure 5*). *er erl1 erl2* cotyledon epidermis develops stomatal clusters (*Figure 5A*; *Video 2*) (*Shpak et al., 2005*). Strikingly, the expression of *MUTEpro::ERL1-YFP* in *er erl1 erl2* conferred massive clusters of arrested meristemoids in the leaf epidermis (*Figure 5B*; *Video 3*). Consistently, stomatal index (SI=total number of stomata/total number of epidermal cells x100) underwent 14-fold reduction from 42.6 ± 5.3% to 2.9 ± 2.1% (mean ± standard deviation) (*Figure 5C*).

To test whether the meristemoid arrests are caused by the ERL1 signaling activated by the endogenous EPF1 peptides, we introduced *MUTEpro::ERL1-YFP er erl1 erl2* into a mutant lacking functional *EPF1*. The *epf1 epf2* double mutant was chosen in order to circumvent the complex effects of EPF-family peptides (*Figure 5D,E*). Indeed, *MUTEpro::ERL1-YFP* in *er erl1 erl2 epf1 epf2* quintuple mutant background was not able to suppress stomatal differentiation, resulting in increased stomatal index (SI) (*Figure 5C,E*). The results suggest that the endogenous EPF peptides are required to elicit the inhibition of stomatal differentiation potential. However, the additional *epf1 epf2* mutation did not fully revert the stomatal differentiation (*Figure 5C*), implying the possibility of further redundancy by *EPF/EPFL* genes.

The EPF-family peptide Stomagen positively regulates stomatal development via directly competing with EPF2 for the binding to ERECTA receptor (*Kondo et al., 2010*; *Sugano et al., 2010*; *Lee et al., 2015*). The Stomagen peptide also associates with ERL1 *in planta* (*Lee et al., 2015*). This prompted us to further test whether the excessive amounts of Stomagen could revert the fate of arrested meristemoids in *MUTEpro::ERL1-YFP* in *er erl1 erl2*. As control, mock treated leaf epidermis showed meristemoid arrests (*Figure 5F*). By contrast, application of bioactive Stomagen peptide (5 µM) to *MUTEpro::ERL1-YFP er erl1 erl2* seedlings conferred clusters of mature stomata (*Figure 5G*). Therefore, Stomagen counteracts the EPF1-ERL1 autocrine signaling, likely via competing with EPF1 for the ERL1 binding. TMM forms a heteromer with ERL1 and is required for EPF1-ERL1 mediated signaling (*Hara et al., 2007*; *Lee et al., 2012*). Consistently, loss-of-function *tmm* mutation fully reverted the arrested meristemoid phenotype of *MUTEpro::ERL1-YFP* in *er erl1 erl2* (*Figure 5C, H, I*). In all these cases YFP signals are detected in a subset of arrested meristemoids, confirming that the restoration of stomatal differentiation by *epf1 epf2*, Stomagen application, and *tmm* mutation is not due to the reduced expression of *MUTEpro::ERL1-YFP* (*Figure 5E–I*). On the basis of these findings, we conclude that the co-presence of EPF1, ERL1, and MUTE in the late meristemoids elicits autocrine inhibitory signaling, leading to a termination of stomatal fate.

It is well known that EPF1 signaling enforces asymmetric spacing division (*Hara et al., 2007*). To further address if the presence of the receptors in stomatal lineage ground cells (SLGCs) ameliorate the autocrine inhibition, we introduced *MUTEpro::ERL1-YFP* in seedlings that harbor wild-type copies of the three *ERECTA*-family genes. Indeed, the expression of *MUTEpro::ERL1-YFP* did not confer discernable phenotype (*Figure 5—figure supplement 1*). Our findings suggest that both a meristemoid and its neighboring cell share the same source of EPF1 signaling peptides, and a disproportionate balance of receptor pools between these two cell types evoke the self-inhibition of stomatal fate.

## Loss of the autocrine EPF1-ERL1 signaling results in high MUTE accumulation and accelerated, dysregulated stomatal differentiation

We have demonstrated that co-presence of EPF1, ERL1 and MUTE in the late meristemoids confers the developmental arrests of stomatal precursors (*Figure 5*). The important question, however, is the role of the EPF1-ERL1 autocrine inhibitory signaling during normal stomatal development. To address this question, a quantitative analysis of MUTE protein accumulation was performed in the presence or absence of TMM, which is required for the EPF1-ERL1 signal transduction. Specifically,

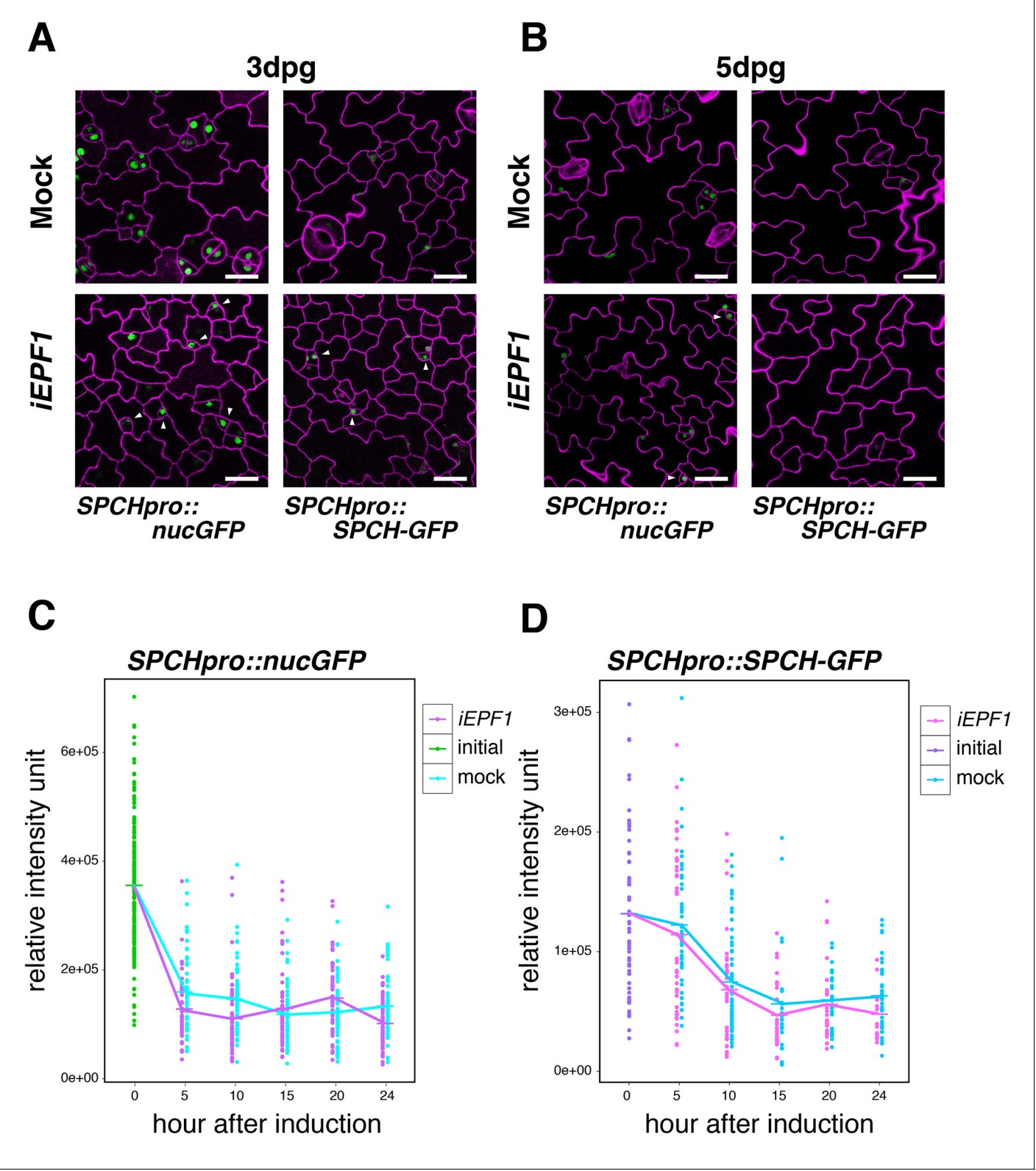

**Figure 4.** Induced *EPF1* overexpression has no effect in *SPCH* promoter activity and SPCH protein levels. (**A**) Confocal microscopy images of abaxial cotyledon epidermis from 3-day-old estradiol-inducible EPF1 seedlings expressing *SPCHpro::nucGFP* (left) and *SPCHpro::SPCH-GFP* (right) that were either mock treated (top) or treated with estradiol (bottom) from germination for induced *EPF1* overexpression (*iEPF1*). Both *SPCH* promoter activity and SPCH-GFP protein are robustly detected in arrested meristemoids (arrowheads) by *iEPF1* at 3-days post germination (dpg). F1 seedlings were used

*Figure 4 continued on next page*

*Figure 4 continued*

for the analysis. Scale bars, 20 µm. Experiments were repeated three times. Three seedlings were analyzed each time. (**B**) Confocal microscopy images of abaxial cotyledon epidermis from 5-day-old estradiol-inducible EPF1 seedlings expressing *SPCHpro::nucGFP* (left) and *SPCHpro::SPCH-GFP* (right) that were either mock treated (top) or treated with estradiol (bottom) from germination for induced *EPF1* overexpression (*iEPF1*). At 5-days post germination (dpg), SPCH-GFP proteins have diminished, reflecting the stomatal cell-lineages have past the early stage. F1 seedlings were used for the analysis. Scale bars, 20 µm. Experiments were repeated three times. Three seedlings were analyzed each time. (**C**) Quantitative analysis of *SPCHpro::nucGFP* levels after *iEPF1* induction at the time indicated on 3-day-old seedlings. Each dot represents the total intensity value of GFP signals from each nucleus expressing *SPCHpro::nucGFP*. To fully cover the entire nuclei expressing GFP, serial Z-stack projection images were used for quantitative analysis (see Materials and methods). Mean value at each time point is connected by the line to visualize the average intensity change over time. Green, initial values at time point 0; Cyan, mock treatment; Purple, *iEPF1* induction. Experiments were repeated three times; n = 6 for time point 0; n = 3 per subsequent time point. Total of 790 nuclei were analyzed. (**D**) Quantitative analysis of SPCH-GFP levels after *iEPF1* induction at the time indicated on 3-day-old seedlings. Each dot represents the total intensity value of GFP signals from each nucleus expressing SPCHpro::SPCH-GFP. To fully cover the entire nuclei expressing SPCH-GFP, serial Z-stack projection images were used for quantitative analysis (see Materials and methods). Mean value at each time point is connected by the line to visualize the average intensity change over time. Purple, initial values at time point 0; Cyan, mock treatment; Magenta, *iEPF1* induction. Experiments were repeated three times; n = 6 for time point 0; n = 3 per subsequent time point. Total of 403 nuclei were analyzed.

we compared the *MUTEpro::MUTE-tagRFP* signals in the nuclei from the *erl1* null mutant background fully rescued by *ERL1pro::ERL1-YFP* (hereafter referred to as 'wild-type (WT)') and those from the *erl1 tmm* double null mutant background rescued by *ERL1pro::ERL1-YFP* (referred to as '*tmm*') (*Figure 6*). The *ERL1pro::ERL1-YFP* transgene was introduced into *tmm* via genetic crosses. As shown in *Figure 6A and C*, MUTE-tagRFP signal intensity in each nucleus is substantially higher in *tmm*, with over 2.5-fold higher average total intensity in each nucleus compared to wild type. The result supports the hypothesis that endogenous autocrine EPF1-ERL1 signaling in the meristemoids prevents excessive MUTE protein accumulation. Also, compared to the wild-type epidermis, more cells express MUTE protein in *tmm*, consistent with the increase in stomatal precursor cells by *tmm* mutation (*Figure 6C*).

Next, we investigated the developmental consequence of excessive MUTE protein accumulation due to the loss of EPF1-ERL1 autocrine signaling. For this purpose, time-lapse live imaging of wild-type and *tmm* seedlings co-expressing *MUTEpro::MUTE-tagRFP* and *ERL1pro::ERL1-YFP* during stomatal development was performed (*Figure 6D,E*; *Videos 4* and *5*). Notably, the duration of proliferation-to-differentiation transition (i.e. hours from the last asymmetric division to the onset of symmetric division of a given stomatal precursor cell) is significantly shorter in *tmm*, with averages of 18.9 hr in wild type (n = 18) vs. 14 hr in *tmm* (n = 28) (*Figure 6B*). Moreover, we observed instances in *tmm* whereby the meristemoid accumulating high MUTE-tagRFP signals undergoes a symmetric division, giving rise to two daughter cells, both expressing MUTE-tagRFP, and eventually differentiating into paired stomata (*Figure 6E*, *Video 5*). It is well known that TMM is required for the asymmetric spacing division of a neighboring stomatal lineage ground cell (SLGC) (*Nadeau and Sack, 2002*; *Lau and Bergmann, 2012*; *Han and Torii, 2016*). On the contrary, the role of TMM in suppressing the symmetric division of a meristemoid has been overlooked. Taken together, our data suggest that loss of the autocrine EPF1-ERL1 signaling during normal stomatal development causes accelerated and dysregulated stomatal differentiation, likely as a ramification of excessive MUTE accumulation.

## Discussion

The EPF-family of secreted cysteine-rich peptides regulates multiple aspects of stomatal development and other developmental processes through the ERECTA-family receptors (*Torii, 2012*; *Lee and De Smet, 2016*). In this study, we discovered an unexpected role for EPF1 and ERL1 in the autocrine regulation of stomatal differentiation potential. The key framework is a negative feedback loop between MUTE and ERL1 whereby MUTE directly upregulates *ERL1* expression (*Figure 2*), and in turn, ERL1 mediates EPF1-peptide signaling leading to inhibition of MUTE activity (*Figure 3*). *ERL1* is also induced by SPCH in earlier stomatal-lineage cells, including MMCs, early meristemoids, and SLGCs (*Figure 1*) (*Lau et al., 2014*). While *EPF1* expression is not regulated by MUTE, *EPF1* turns on during meristemoid-to-GMC transition (*Figure 2*). Based on these findings, we proposed a

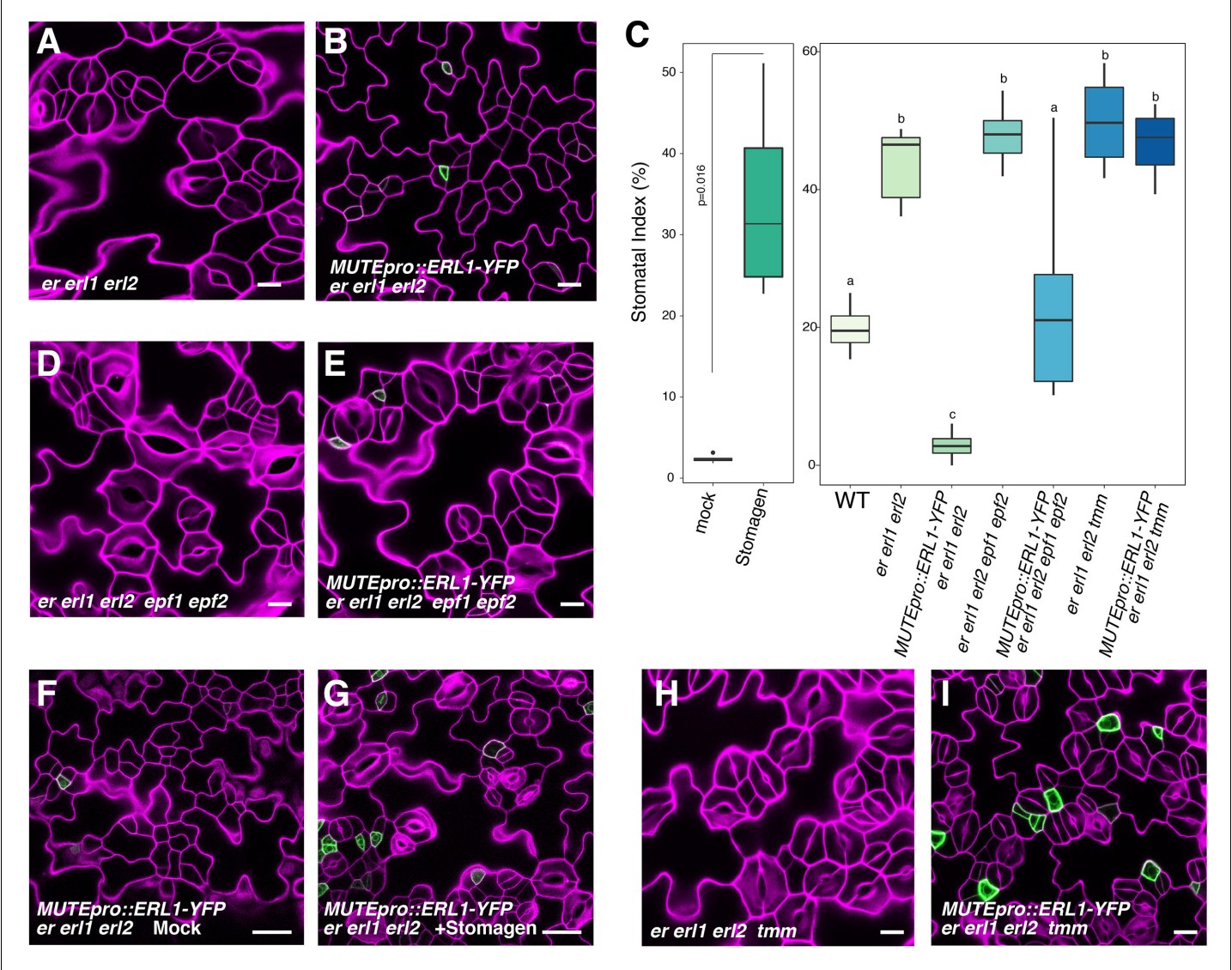

**Figure 5.** Absolute co-expression of ERL1 and MUTE confers meristemoid arrests. Shown are confocal microscopy images of cotyledon abaxial epidermis from 7-day-old seedlings of the following genotypes: (A) *er erl1 erl2*; (B) *MUTEpro::ERL1-YFP* in *er erl1 erl2*; (D) *er erl1 erl2 epf1 epf2*; (E) *MUTEpro::ERL1-YFP* in *er erl1 erl2 epf1 epf2*; (F) *MUTEpro::ERL1-YFP* in *er erl1 erl2* mock treated; (G) *MUTEpro::ERL1-YFP* in *er erl1 erl2* treated with 5 μM Stomagen peptide; (H) *er erl1 erl2 tmm*; (I) *MUTEpro::ERL1-YFP* in *er erl1 erl2 tmm*. T1 transgenic seedlings of *MUTEpro::ERL1-YFP er erl1 erl2*; *MUTEpro::ERL1-YFP er erl1 erl2 epf1 epf2*; and *MUTEpro::ERL1-YFP er erl1 erl2 tmm* were used for the analysis. T2 seedlings of *MUTEpro::ERL1-YFP er erl1 erl2* were used for the mock or Stomagen treatment. Scale bars, 10 μm (A, B, D, E, H, I), 25 μm (F, G). (C) Quantitative analysis. Stomatal index (SI) of the cotyledon abaxial epidermis from 7-day-old seedlings of respective genotypes. For each genotype, images from six seedlings were analyzed. Welch's Two Sample T-test was performed for mock vs. Stomagen application (Left). One-way ANOVA followed by Tukey's HSD test was performed for comparing all other genotypes and classify their phenotypes into three categories (a, b, and c).

The following figure supplement is available for figure 5:

**Figure supplement 1.** *MUTEpro::ERL1-YFP* does not cause meristemoid arrest in the presence of functional *ERECTA*-family genes.

coincidence model where the overlapping expression of EPF1, ERL1, and MUTE during a narrow developmental window in the late meristemoids triggers downregulation of MUTE (*Figure 7*), thereby ensuring timely transition to stomatal differentiation mediated by FAMA.

The co-expression of EPF1 and ERL1 triggering autocrine signaling challenges a traditional view of plant secreted peptide hormones as intercellular signals (*Murphy et al., 2012*). Known EPF/EPFL

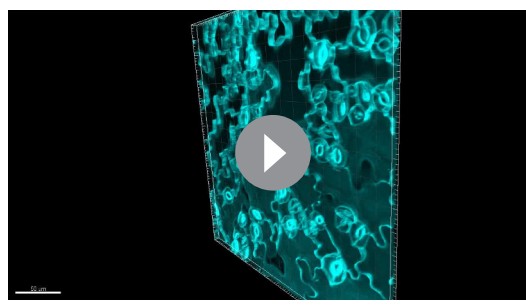

**Video 2.** Three-dimensional projection of the *er erl1 erl2* phenotype. Shown is a projection of z-stack images of *er erl1 erl2* from the abaxial cotyledon epidermis of the 7-day-old seedling. Scale bar, 50 μm.

peptides function in a non-cell autonomous manner. For example, EPFL9/Stomagen secreted from internal mesophyll tissues is perceived by ERECTA in the protoderm (*Lee et al., 2015*). EPFL4 and EPFL6 are inter cell-layer signals between the stem endodermis and phloem to promote inflorescence elongation via ERECTA (*Uchida et al., 2012*). Recently, we have reported that EPFL2 peptide promotes leaf serration as a ligand for ERECTA-family (*Tameshige et al., 2016*). Interestingly, EPFL2 and its primary receptor, ERL2, exhibit inverse expression patterns: *ERL2* shows the highest expression at the leaf teeth tips where *EPFL2* expression is excluded. Therefore, EPF/EPFL-family peptides mediate signals both in autocrine and paracrine manners.

Both *epf1* mutation and a dominant-negative ERL1 receptor confer a violation of one-cell spacing rule (*Hara et al., 2007*; *Lee et al., 2012*). Based on this phenotype, a paracrine model was proposed initially, whereby a signal (EPF1) emanating from the meristemoids/GMCs will be perceived by receptors (ERL1 and TMM) in the neighboring stomatal lineage ground cell (SLGC) to orient asymmetric spacing division (*Figure 7A*). However, based on the high expression of EPF1, ERL1, and TMM in the meristemoids/GMCs, an alternative autocrine model has been described (*Figure 7B*) (*Facette and Smith, 2012*). Here, EPF1 peptide is perceived by ERL1/TMM in the meristemoids/GMCs. A subsequent downstream signal transduction promotes the expression of a yet unidentified ligand X, which will be secreted and perceived by a yet unidentified receptor Y on the surface of neighboring stomatal lineage ground cells (SLGCs). This eventually leads to the oriented asymmetric spacing division (*Figure 7B*) (*Facette and Smith, 2012*). Our findings revise these models and propose a buffered system, whereby the pool of EPF1 is shared by both the meristemoid and its neighboring stomatal lineage ground cell (SLGC) (*Figure 7C*). ERL1 and TMM also accumulates at the plasma membrane of stomatal lineage ground cells (SLGCs), albeit in lesser amounts (*Figure 1*) (*Nadeau and Sack, 2002*; *Horst et al., 2015*). ERL1 in the stomatal lineage ground cells (SLGCs) most likely perceives the EPF1 peptides emitted from neighboring meristemoids, and this EPF1-ERL1 paracrine signaling orients the asymmetric spacing divisions. Based on this model (*Figure 7C*), the paracrine action of EPF1 also limits the inhibitory activity of EPF1-ERL1 autocrine signals in the meristemoids, since ERL1 in the meristemoids and in neighboring stomatal lineage ground cells (SLGCs) both share the same source of EPF1 signals.

It is important to note that the absolute co-expression of ERL1 and MUTE conferred severe meristemoid arrests due to excessive activation of ERL1 signaling pathway, when all three endogenous ERECTA-family receptors are missing (*Figure 5*). Indeed, *MUTEpro::ERL1-YFP* does not confer any discernable phenotype if the wild-type copies of the three *ERECTA*-family genes are present (*Figure 5—figure supplement 1*). It is therefore most likely that, in addition to ERL1, ERECTA and ERL2 receptor population expressed in stomatal lineage ground cells (SLGCs) buffer excessive EPF1 peptide ligands to ensure the proper and adequate level of MUTE inhibition. The idea of signal buffering by sister ERECTA-family receptors is supported by their known partial redundancy with complex genetic interactions (*Shpak et al.,*

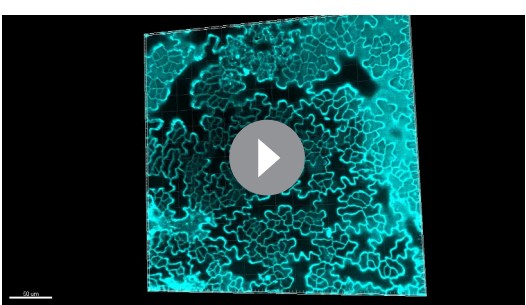

**Video 3.** Three-dimensional projection of the *MUTEpro::ERL1-YFP er erl1 erl2* phenotype. Shown is a projection of z-stack images of *MUTEpro::ERL1-YFP* in *er erl1 erl2* from the abaxial cotyledon epidermis of the 7-day-old T1 seedling. Numerous arrested meristemoids are visible with no stomatal differentiation. Scale bar, 50 μm.

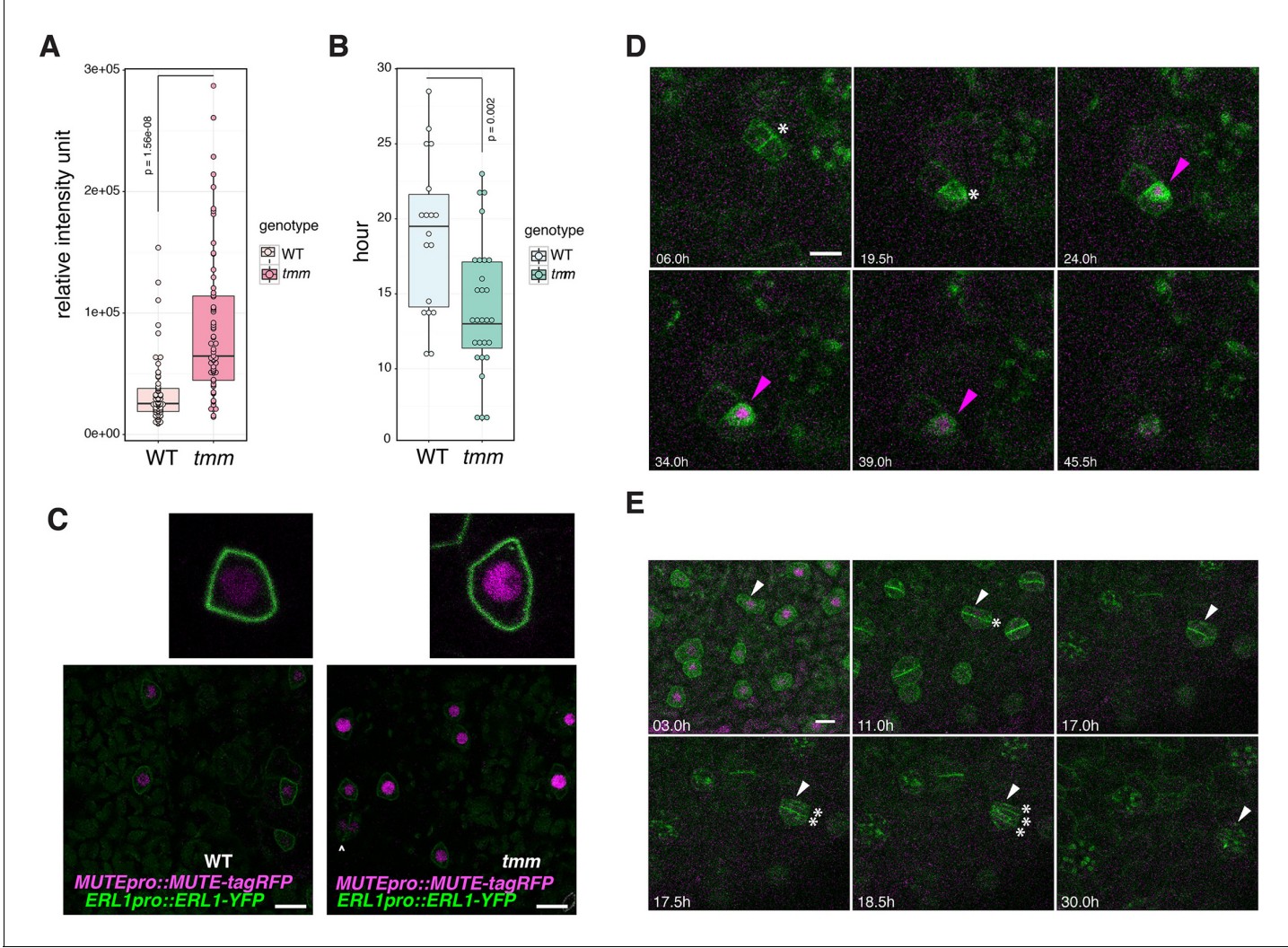

**Figure 6.** Perturbed EPF1 signaling results in higher amount of MUTE protein, accelerated stomatal differentiation, and symmetric division of GMC. (**A**) Quantitative analysis of RFP signal intensity from the nuclei expressing *MUTEpro::MUTE-tagRFP*. WT and *tmm* refers to *MUTEpro::MUTE-tagRFP ERL1pro::ERL1-YFP* in *erl1* and *MUTEpro::MUTE-tagRFP ERL1pro::ERL1-YFP* in *erl1 tmm*, respectively. F3 seedlings were used for the analysis. Experiments were repeated three times. Total numbers of nuclei analyzed; n = 120 (WT); n = 135 (*tmm*). Welch's Two Sample T-test was performed. (**B**) Quantitative analysis of a duration of proliferation-to-differentiation transition during stomatal development (hours from last asymmetric division of a meristemoid to onset of GMC symmetric division). *MUTEpro::MUTE-tagRFP ERL1pro::ERL1-YFP* in *erl1* was referred as WT and *MUTEpro::MUTE-tagRFP ERL1pro::ERL1-YFP* in *erl1 tmm* was referred as *tmm*. F3 seedlings were used for the analysis. n = 18 (WT); n = 28 (*tmm*). Welch's Two Sample T-test was performed. (**C**) Z-stack confocal microscopy image projections of true leaf abaxial epidermis from 7-day-old F3 seedlings expressing *MUTEpro::MUTE-tagRFP ERL1pro::ERL1-YFP* in *erl1* ('WT'); and *MUTEpro::MUTE-tagRFP ERL1pro::ERL1-YFP* in *erl1 tmm* ('*tmm*'). For quantitative comparison, image acquisition and processing were performed using the identical settings. Scale bars, 10 μm. Top insets, a representative late meristemoid from 'WT' (left) and '*tmm*' (right). (**D**) Time-lapse imaging of stomatal differentiation in F3 seedlings of 'wild type (WT)'; *ERL1pro::ERL1-YFP* and *MUTEpro::MUTE-tagRFP* in *erl1*. The site of asymmetric entry and amplifying divisions are indicated in white asterisks. The MUTE protein accumulates in the nucleus of the late meristeoid and disappears as the GMC differentiates (magenta arrowheads). Time points after image collection are indicated in hours. See accompanying *Video 4*. Scale bar, 10 μm. Experiments were repeated three times. Total seedlings analyzed; n = 9. (**E**) Time-lapse imaging of stomatal differentiation in F3 seedlings of '*tmm*'; *ERL1pro::ERL1-YFP* and *MUTEpro::MUTE-tagRFP* in *erl1 tmm*. Occasionally, a late meristemoid accumulating MUTE-tagRFP (white arrowhead) maintain MUTE accumulation after initial GMC symmetric division (*; in 11.0 hr), leading to an extra round of symmetric division (asterisks in 17.5 hr and 18.5 hr). Time points after image collection are indicated in hours. See accompanying *Video 5*. Scale bar, 10 μm. Experiments were repeated three times. Total seedlings analyzed; n = 9.

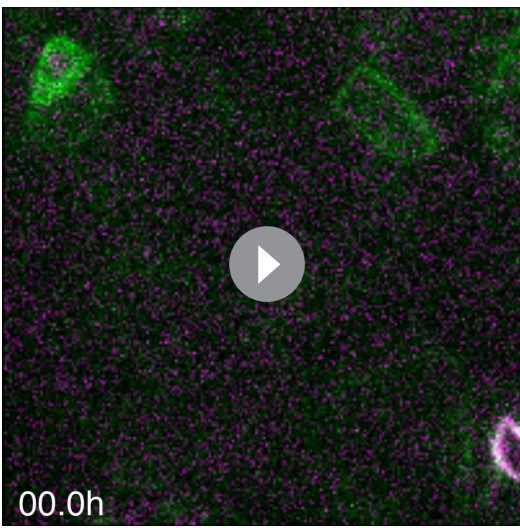

**Video 4.** Time-lapse movie of normal stomatal differentiation from the line co-expressing *ERL1pro:: ERL1-YFP* and *MUTEpro::MUTE-tagRFP*. Projection of z-stack images of *ERL1pro::ERL1-YFP* and *MUTEpro:: MUTE-TagRFP* from the abaxial cotyledon epidermis of an 1-day-old *erl1* seedling were shown in 30 min time interval. Time points after image collection are indicated in hours. F3 seedlings were used for the analysis.

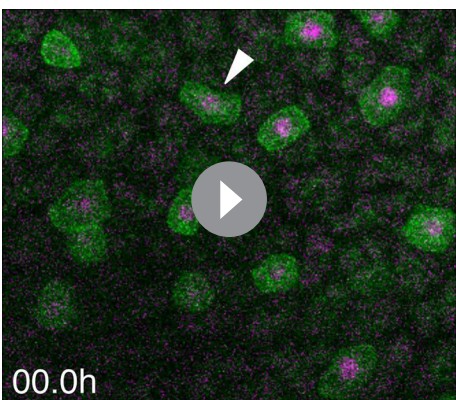

**Video 5.** Time-lapse movie of abnormal stomatal differentiation from *tmm* mutant line co-expressing *ERL1pro::ERL1-YFP* and *MUTEpro::MUTE-tagRFP*. Projection of z-stack images of *ERL1pro::ERL1-YFP* and *MUTEpro::MUTE-TagRFP* from the abaxial cotyledon epidermis of an 1-day-old *erl1 tmm* seedling were shown in 30 min time interval. Time points after image collection are indicated in hours. The arrowhead points to one cell, which develops into two parallel-aligned stomata due to an extra round of symmetric division. F3 seedlings were used for the analysis.

*2004*, *2005*). Moreover, EPFL4/6 (CHAL/CLL1), which normally act as ligands for ERECTA-mediated inflorescence elongation, interfere with this buffering mechanism and confer stomatal clustering in the stems the absence of TMM (*Abrash and Bergmann, 2010*; *Abrash et al., 2011*; *Uchida et al., 2012*). Very recently, a VST-family of membrane-associated proteins was reported as a modulator of ERECTA-family signaling, via specifically interacting with ERL2 (*Ho et al., 2016*). Such factors may contribute to fine-tuning the EPF1-ERL1 signaling.

Our work also revealed that, while two paralogous ligand-receptor pairs, EPF2-ERECTA and EPF1-ERL1, enforce stomatal development via acting on the core stomatal bHLH proteins, the architectures of their regulatory framework are different. EPF2, but not its main receptor ERECTA, is the direct SPCH target (*Lau et al., 2014*; *Horst et al., 2015*). Conversely, ERL1, but not its main ligand EPF1, is the direct MUTE target (*Figure 2*). The biochemical mechanism of MUTE downregulation is unknown, but lines of evidence suggest the involvement of a MAP kinase cascade. First, YODA, MKK4/5, and MPK3/6 are genetically acting downstream of the three ERECTA-family RKs (*Lampard et al., 2008*; *Bemis et al., 2013*). Second, the expression of constitutively-active MKK4/5 by the *MUTE* promoter led to arrested meristemoids (*Lampard et al., 2009*), thus phenocopying EPF1 peptide application or overexpression (*Hara et al., 2007*; *Lee et al., 2012*). Unlike SPCH, however, MUTE does not possess a predicted MAP kinase target domain (*Davies and Bergmann, 2014*), raising a question of whether MUTE is a MPK3/6 substrate. In any event, functional specialization and re-wiring of regulatory circuits among the paralogs of EPF peptides, ERECTA-family receptor kinases, and stomatal bHLH transcription factors highlight the evolutionary mechanism underlying the sequential steps of stomatal differentiation steps.

Intrinsic polarity of meristemoids and asymmetric daughter cell fates are regulated by BASL (*Dong et al., 2009*). Recent studies have revealed that polarized BASL in the stomatal lineage ground cells (SLGCs) enhances the MAP kinase activity, resulting in lowered SPCH abundance (*Zhang et al., 2015*, *2016*). In the absence of *EPF1*, GFP-BASL occasionally exhibits improper polar localization to the 'wrong' site within a periphery of the stomatal lineage ground cells (SLGCs), which predicts the mis-oriented site of asymmetric spacing division (*Dong et al., 2009*). It would be an interesting

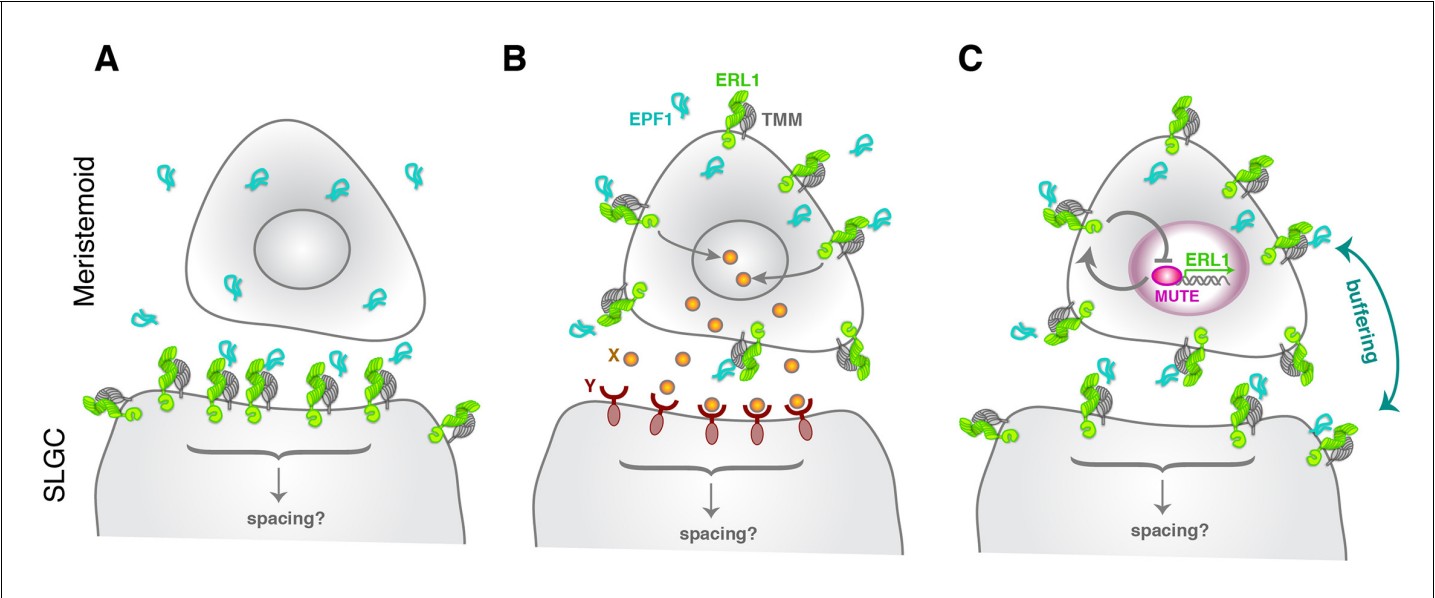

**Figure 7.** Model diagram of how EPF1-ERL1 peptide-receptor signaling regulates stomatal differentiation. (**A**) Paracrine signaling model: Here, EPF1 peptide (cyan) emanating from a meristemoid is perceived by ERL1 (light green) and TMM (gray) localized at the plasma membrane of a neighboring stomatal lineage ground cell (SLGC). The activated signaling will in turn regulate asymmetric spacing division. (**B**) Autocrine signaling model: *Facette and Smith (2012)* proposed an alternate, autocrine signaling model based on the observations that *EPF1*, *ERL1*, and *TMM* all exhibit the highest expression in the meristemoid (*Nadeau and Sack, 2002*; *Shpak et al., 2005*; *Hara et al., 2007*). Here, EPF1 secreted from a meristemoid is perceived by ERL1-TMM on a surface of the same meristemoid. The downstream signaling leads to expression of a yet unidentified peptide (X: orange), which will be perceived by a yet unidentified receptor (Y: brown) on a surface of the neighboring stomatal lineage ground cell (SLGC). This triggers signal transduction regulating asymmetric spacing division. (**C**) Model based on the current study: Both *ERL1* and *TMM* are expressed in a meristemoid and a stomatal lineage ground cell (SLGC), since they are both upregulated by SPCH at an earlier stage (*Lau et al., 2014*; *Horst et al., 2015*). During the meristemoid-to-GMC transition, MUTE upregulates *ERL1* expression and this causes high accumulation of ERL1 in the meristemoid over the stomatal lineage ground cell (SLGC). Meanwhile, EPF1 is expressed and secreted from the late meristemoid by an independent mechanism. A perception of EPF1 by the meristemoid-expressed ERL1 and TMM triggers signal transduction leading to downregulation of MUTE, thereby restricting MUTE activity via an autocrine mechanism. The same pool of EPF1 is also perceived by ERL1 and TMM expressed in the neighboring stomatal lineage ground cell (SLGC), which trigger downstream signal transduction to ensure asymmetric spacing division via a paracrine mechanism. Consequently, receptors in the meristemoids and stomatal lineage ground cells (SLGCs) buffer the pool of EPF1 ligands. Note that our model does not exclude the possibility that yet another unknown signals (like X and Y) operate simultaneously during the stomatal differentiation process.

future question to address how the EPF1-ERL1 signaling pathway influences the location of BASL polarization and stomatal-lineage cell polarity.

It has become evident that plants make use of a myriad of peptide signals to regulate diverse developmental processes. Accumulating evidence suggest that plant peptide signals mediate both local cell-cell communications and long-distance root-to-shoot signaling (*Brand et al., 2000*; *Searle et al., 2003*; *Hirakawa et al., 2008*; *Okamoto et al., 2013*; *Tabata et al., 2014*). In most cases, however, the exact source and recipient cells of signals remains unclear. Previously studies have identified ligand-receptor systems that share the same expression domains. For instance, the tomato cysteine-rich secreted protein LAT52 and its receptor LRR-RK, LePRK2, are both expressed in pollen (*Muschietti et al., 1998*; *Tang et al., 2002*). It was speculated that pollen may monitor its germination state via LAT52-LePRKs autocrine signaling (*Tang et al., 2002*). Another example is CLE45-BAM3, a small secreted peptide ligand-LRR-RK pair: both CLE45 and BAM3 are expressed in protophloem tissue in roots, and their signal transduction leads to the inhibition of protophloem differentiation (*Depuydt et al., 2013*; *Rodriguez-Villalon et al., 2014*). It is not yet known whether these ligand-receptor signaling systems constitute feedback regulation with target transcription factors. In any event, the autocrine signaling may be a prevalent mode of peptide-receptor kinase signaling in plants. Here, focusing on stomatal development, where the differentiation potential of bipotent precursor cells can be monitored at a single cell resolution, one can decipher the contribution of autocrine peptide signaling and the buffering of autocrine vs. paracrine signals between two

sister cells with different developmental fate. In animals, both paracrine and autocrine peptide signaling contributes to stem cell proliferation. For instance, Wnt, a key secreted cell-cell signal during animal development (*Logan and Nusse, 2004*), participates in the autocrine regulation of epidermal stem cell renewal in mice (*Lim et al., 2013*). A versatile use of peptide ligands for the maintenance and differentiation of adult stem cells may highlight yet another theme in the logic of development in multicellular organisms.

## Materials and methods

### Plant materials and growth condition

*Arabidopsis thaliana* Columbia (Col) accession was used as a wild type. The following mutants and reporter lines were reported previously: *erl1-2, er-105 erl1-2 erl2-1* (*Shpak et al., 2005*); *EPF1pro:: erGFP, epf1,* and *tmm-KO* (*Hara et al., 2007*); *mute, MUTEpro::MUTE-GFP, SPCHpro::SPCH-GFP* (*Pillitteri et al., 2007*); *scrm-D, SCRMpro::GFP-SCRM* (*Kanaoka et al., 2008*); *epf1 epf2 er erl1 erl2* (*Hara et al., 2009*); *ERL1pro::ERL1-YFP, Est::EPF1 (iEPF1)* (*Lee et al., 2012*); and *SCRMpro::nucGFP* and *SPCHpro::nucGFP* (*Horst et al., 2015*). Plants were grown under the long-day condition.

### Plasmid construction and generation of transgenic plants

The following plasmids were generated for this study: pAR202 (*MUTE* promoter), pAR205 (*MUTE* genomic sequence with no stop codon), pJSL105 (*ERL1* genomic sequence with no stop codon), pBK5 (*MUTEpro::ERL1-YFP*), pJT163 (*MUTEpro::MUTE-tagRFP*), pLJP245 (*Est::MUTE*), and *MUTE-pro::nucYFP*. To construct pBK5, three-way Gateway recombination was performed using pAR202, pJSL105, and R4L1pGWB540 (*Nakagawa et al., 2008*). To construct pJT163, three-way Gateway recombination was performed using pAR202, pAR205, and R4L1pGWB659 (*Nakagawa et al., 2008*). pGWB vectors are kind gift from Prof. Tsuyoshi Nakagawa (Shimane Univ.). For detailed information about the plasmids, see *Supplementary file 1*. Plasmids are transformed into *Agrobacterium* GV3101/pMP90 and subsequently to *Arabidopsis* by floral dipping. Over 10 lines were characterized for the phenotypes and reporter gene expressions. Established reporter lines were introduced into higher-order mutant backgrounds or dual reporter backgrounds via genetic crosses. The genetic crosses made in this study include: *MUTEpro::MUTE-tagRFP x ERL1pro::ERL1-YFP erl1; MUTEpro:: MUTE-tagRFP x EPF1pro::erGFP; MUTEpro::MUTE-GFP mute x MUTEpro::MUTE-tagRFP; Est::EPF1 x MUTEpro::nucYFP; Est::EPF1 x MUTEpro::MUTE-GFP mute; Est::EPF1 x SCRMpro::nucGFP; Est:: EPF1 x SCRMpro::GFP-SCRM; Est::EPF1 x SPCHpro::nucGFP; Est::EPF1 x SPCHpro::SPCH-GFP spch; er erl1 erl2 x tmm; MUTEpro::ERL1-YFP er erl1 erl2 x Col-0 wild type; ERL1pro::ERL1-YFP erl1 tmm x MUTEpro::MUTE-tagRFP.* The following transgenic lines in higher-order mutants were generated by floral dipping: *ERL1pro::ERL1-YFP* into *er erl1 erl2; MUTEpro::ERL1-YFP* into *er erl1 erl2; MUTEpro::ERL1-YFP* into *epf1 epf2 er erl1 erl2; MUTEpro::ERL1-YFP* into *er erl1 erl2 tmm*. For primer DNA sequence used for plasmid construction, see *Supplementary file 2*. Estradiol induction was performed as described previously (*Lee et al., 2012*).

### Peptide expression, purification, refolding, and application

EPF1 peptides were expressed, purified, and refolded as previously described (*Lee et al., 2012, 2015*). Stomagen peptide was custom synthesized (BioSynthesis, Lewisville, TX) and subjected to refolding. For quality control, the refolded peptides were analyzed by High Pressure Liquid Chromatography (Hewlett Packard 1100 Series) using ZORBAX 300 Extended-C18 column (Agilent), and each major peak was analyzed by Esquire-LC ion-trap mass spectrometer (Bruker Daltonics, Billerica, MA). The bioactivity of each refolded peptide batch was verified by seedling bioassays (*Lee et al., 2015*) prior to performing any experiments. For each peptide or mock treatment, at least three biological repeats were performed.

### qRT-PCR

Homozygous transgenic lines for iMUTE were grown on ½ MS agar media for 4 days. 40–50 seedlings for each replicate were bathed in a ½ MS liquid media containing 10 µM estradiol or DMSO (mock). Total RNA was isolated from the seedlings using RNeasy Plant Mini Kit (Qiagen, 74904) and DNase I digestion (Qiagen, 79254) was performed on column during the RNA extraction according

to the instructions of the manufacturer. 1 µg of RNA was converted to cDNA using iScript cDNA synthesis kit (Bio-Rad, 1708891) according to the instructions of the manufacturer. First-strand cDNA was diluted 1:7 in double distilled water and used as template for Real-time qPCR. qRT-PCR was performed as described below (see Chromatin immunoprecipitation section). At least four 10-fold dilution series were amplified by the same primers to set standard curve and calculate the quantity of the gene expression. Relative expression was calculated by dividing *ACT2* gene expression over the specific-gene expression. Three biological replicates were performed. See *Supplementary file 2* for oligo DNA primer sequence used for qRT-PCR.

## Chromatin Immunoprecipitation

160 mg seeds were sown on ½ MS agar media and seedlings were collected at 42 ~ 48 hr after germination in liquid nitrogen, which yields ca. 1.0 g of fresh weight. Frozen tissue was ground to fine powder and re-suspended in 20 ml of ChIP extraction buffer 1 (*Bowler et al., 2004*) containing 1% formaldehyde (Sigma, F1635). Samples were incubated at 4°C for 10 min for DNA-protein cross-linking. To quench the cross-linking 2 M glycine was added, then samples were incubated at 4°C for 5 min (*Cui et al., 2016*). The chromatin was extracted twice in ChIP extraction buffer 2 (*Bowler et al., 2004*). Isolated chromatin was lysed in 100 µl of Nuclei Lysis buffer on ice for 30 min with occasional stirring as described in *Yamaguchi et al. (2014)*. Subsequently, volumes were brought up to 300 µl with ChIP dilution buffer with Triton X-100 (*Yamaguchi et al., 2014*). Sonication was performed to shear DNA using Bioruptor Plus UCD-300 sonicator (Diagenode), 30 s on/off cycles for 15 times at High power setting. After the sonication, samples were diluted up to 1 ml with ChIP dilution buffer with Triton and spun at full speed. The supernatant was precleared using Dynabeads protein G (Invitrogen, 10004D) for 1 hr at 4°C. Pre-cleared solution was subjected to overnight immunoprecipitation at 4°C. 2 µl of anti-GFP antibody (Ab290, Lot: GR278073-1) was added in 1 ml pre-cleared chromatin sample. Subsequent steps (capturing, washing, reverse cross-linking of immune-complex and DNA purification) were performed as described in *Yamaguchi et al. (2014)*. Quantitative PCR (qPCR) was carried out using gene specific primers (*Supplementary file 2*). The qPCR was run using iTaq Universal SYBR Green Supermix on CFX96 real-time system (Bio-Rad, 1725121) with the following conditions: 95°C for 1 min; 45 cycles of 95°C for 5 s, 60°C for 30 s (Cycling Stage), 65–95°C 0.5°C increments at 5 s/step (Melting Curve Stage). At least three technical replicates were used for ChIP qPCR. Three biological replicates were performed. A dilution series of input was amplified simultaneously using the same primer pairs to calculate the quantity of the ChIPed DNA as % input (*Yamaguchi et al., 2014*).

## Confocal microscopy and time-lapse imaging

The Zeiss LSM700 inverted confocal microscope (Thornwood, NY, USA) and the Leica SP5X-WLL inverted confocal microscope (Solms, Germany) were used for imaging. Time-lapse cotyledon imaging of ERL1-YFP was prepared as described previously with 20x/0.8 Apochromat lens (x1 zoom) on Zeiss LSM700 (*Pillitteri et al., 2011*; *Peterson and Torii, 2012*). For the multicolor images of YFP and RFP, transgenic seedlings were observed with a 63x/1.2 W Corr lens on Leica SP5X-WLL using Hy-D detector. 514 nm laser was used to excite YFP and 560 nm laser was used to excite RFP. The emission filter was 524–550 nm for YFP and 594–637 nm for RFP. For multicolor images of GFP and RFP, transgenic seedlings were observed with a 63x/1.2 W Corr lens on Leica SP5X-WLL using Hy-D detector. The 488 nm laser was used to excite GFP and the 560 nm laser was used to excite RFP. The emission filter was 500–515 nm for GFP and 594–637 nm for RFP. The Z-stack projection images were taken at the interval of 0.99 µm, covering the thickness of the entire cell. For qualitative image presentation, Adobe Photoshop CS6 was used to trim and uniformly adjust the contrast/brightness.

### Quantitative analysis and statistics

The Leica LAS AF software and Imaris ver. 8.1.3 (Bitplane) were used for post-acquisition image processing. Quantitative analysis of MUTE signal intensity was performed using Imaris ver. 8.1.3 briefly as the following. An extensive series of Z-stack confocal images covering the entire meristemoids (~17 layers) were subjected to surface rendering in red or green channel to capture MUTE-tagRFP, MUTE-GFP, SPCH-GFP, GFP-SCRM, SPCHpro::nucGFP, or SCRMpro::nucGFP expressing nuclei in the 3-D space (for each time point for time course). Cut-off value was set as 0.85 for

sphericity, which effectively removed objects with non-specific signals. Intensity sum and standard deviation for each nucleus were subsequently calculated. Statistical analyses were performed using R ver. 3.3.1. Graphs were generated using R ggplot2 package or Microsoft Excel. See *Supplementary file 4* for raw data for R-based analyses and actual R scripts.

## Acknowledgements

We thank Tsuyoshi Nakagawa for the generous gift of Gateway cloning vectors; Lynn Pillitteri, Jin Suk Lee, Bob Kao, and Amanda Rychel for constructing the plasmids; Michal Maes for refolding Stomagen peptide; Kylee Peterson for assistance in genetic crosses and screening transgenic lines; University of Washington Chemistry Mass Spectrometry Facility for the Mass-spec analysis; Aarthi Putarjunan for commenting on the manuscript.

## Additional information

### Funding

| Funder | Grant reference number | Author |
| --- | --- | --- |
| Howard Hughes Medical Institute | | Keiko U Torii |
| Gordon and Betty Moore Foundation | GBMF3035 | Keiko U Torii |
| National Science Foundation | MCB-0855659 | Keiko U Torii |

The funders had no role in study design, data collection and interpretation, or the decision to submit the work for publication.

### Author contributions

XQ, Data curation, Formal analysis, Validation, Investigation, Methodology, Writing—review and editing, Performed all experiments except for ChIP assays and peptide production; S-KH, Data curation, Formal analysis, Investigation, Visualization, Methodology, Writing—review and editing, Performed ChIP assays and qRT-PCR; JHD, Data curation, Prepared bioactive EPF1 peptides and assisted with peptide bioassays, Did stomatal Index analysis; JMG, Resources, Writing—review and editing, Constructed MUTEpro::MUTE-tagRFP construct and transgenic lines; MI, Resources, Constructed MUTEpro::nucYFP, Read and approved the manuscript; AKH, Resources, Data curation, Prepared bioactive EPF1 peptides and assisted with peptide bioassays; KUT, Conceptualization, Data curation, Formal analysis, Funding acquisition, Investigation, Visualization, Writing—original draft, Project administration, Writing—review and editing, Performed genetic crosses

### Author ORCIDs

Xingyun Qi, http://orcid.org/0000-0003-1261-1362
Jonathan H Dang, http://orcid.org/0000-0002-0458-4010
Alex K Hofstetter, http://orcid.org/0000-0002-3709-9783
Keiko U Torii, http://orcid.org/0000-0002-6168-427X

## Additional files

### Supplementary files

• Supplementary file 1. Plasmid constructs generated in this study List of plasmid constructs and selection markers.

• Supplementary file 2. Oligo DNA primers and their sequences used in this study List of oligo DNA names, sequence, and their use.

• Supplementary file 3. Raw data of ChIP experiments Numerical values of the ChIP experiments.

• Supplementary file 4. Raw data and R-based analyses List of original data for stomatal index, qRT-PCR, quantitative data of reporter fluorescent protein accumulation and cell-state transition. Also includes R-scripts written for the analyses and graph presentations.

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
