## [Decision Letter]

Thank you for submitting your article "Autocrine regulation of stomatal differentiation potential by EPF1 and ERECTA-LIKE1 ligand-receptor signaling" for consideration by *eLife*. Your article has been favorably evaluated by Christian Hardtke (Senior Editor) and three reviewers, one of whom, Sheila McCormick (Reviewer #1), is a member of our Board of Reviewing Editors. The following individual involved in review of your submission has agreed to reveal their identity: Eugenia Russinova (Reviewer #2).

The reviewers have discussed the reviews with one another and the Reviewing Editor has drafted this decision to help you prepare a revised submission.

Summary:

This paper deals with the role of the secreted cysteine-rich peptide EPF1 in stomatal development. The authors present data supporting a negative feedback loop between ERL1 and MUTE. MUTE upregulates ERL1 expression and ERL1-mediated EPF1 signaling inhibits MUTE to specify the meristemoid-to-GMC transition switch. They further show that loss of EPF1-ERL1 signaling results in MUTE accumulation and stomatal development defects. The authors used an elegant combination of quantitative microscopy and genetic experiments to support this model. The work is novel, provides a function for EPF1 and is of interest to plant (and other) communities. Nonetheless, some aspects of the work are oversold. The authors need to improve the manuscript before publication in the following aspects:

Essential revisions:

1) Perform expression analyses to show that MUTE is induced after estradiol application and that this induction correlates with the increase and the decrease in expression of ERL1 and EPF1 respectively, hours after induction (Figure 2). The same for the control treatment. Regarding Figure 3 and Figure 4; carry out expression analyses to show that EPF1 is induced after estradiol application. As in Figure 3, they should also quantify the expression of pSCRM-GFP and SCRM-GFP fusion proteins in the nuclei and for the data presented in Figure 3—figure supplement 2 and Figure 4.

2) Provide information as to how the marker lines were generated (transformation or crossing), plant generation used, expression levels of the respective tagged proteins and background used (wild type, mutant). It is not always clear what was the genetic background used to express the tagged proteins, for example in Figure 2, are ERL1-YFP and MUTE-tagRFP expressed in the Col-0 or erl1 background? Similarly for Figure 2.

3) Fluorescently-tagged receptors are not always functional (sometimes they are only partially functional, for example BAK1, FLS2); therefore is important that the authors quantify the phenotypes of the rescued *er erl1 erl2* triple mutant (Figure 5), in order to be more convincing that the ERL1-GFP fusion is functional (similar experiments were done for pMUTE::ERL1-YFP introduced in the *er erl1 erl2* triple mutant). Also, the expression levels should be close to the endogenous, as even when using an endogenous promoter, overexpression of the transgene is frequently observed. Is the MUTE-GFP fusion functional?

4) TagRFP is a slowly maturing protein (100 min, in contrast to 10 min for GFP) and this might affect the co-localization dynamics shown in Figure 2, Figure 5. Provide data or discuss this possibility.

5) It is not clear why MUTE-GFP in the scrm-D background was used for the ChIP assays and not MUTE-GFP in wild type. Is SCRM required for MUTE binding? Explain or discuss.

---

## [Author Response]

*Essential revisions:*

*1) Perform expression analyses to show that MUTE is induced after estradiol application and that this induction correlates with the increase and the decrease in expression of ERL1 and EPF1 respectively, hours after induction (Figure 2).*

This is a valid point. We have performed qRT-PCR to verify the induction of estradiol-inducible *MUTE* overexpression (revised Figure 2). The induction kinetics of *MUTE* precedes that of *ERL1.*

*The same for the control treatment. Regarding Figure 3 and Figure 4; carry out expression analyses to show that EPF1 is induced after estradiol application.*

The RT-PCR of the estradiol-inducible *EPF1* has been published previously (Lee et al., 2012 Genes Dev). In response to the reviewers’ comment, we have further performed qRT-PCR analysis to verify the induction kinetics of estradiol-inducible *EPF1* overexpression in the time frame relevant to our MUTE reporter analysis. The data are now provided as Figure 3—figure supplement 1.

*As in Figure 3, they should also quantify the expression of pSCRM-GFP and SCRM-GFP fusion proteins in the nuclei and for the data presented in Figure 3—figure supplement 2 and Figure 4.*

As requested, we have performed quantitative time-course image analyses of individual nuclei expressing *SCRMpro::nucGFP, SCRMpro::GFP-SCRM, SPCHpro::nucGFP,* and *SPCHpro::SPCH-GFP* upon induced *EPF1* overexpression (*iEPF1*) within 24 hour time frame (see revised Figure 3, and Figure 4). As we described in our original manuscript, 24-hour post induction of *iEPF1* is sufficient to observe the complete disappearance of MUTE-GFP protein accumulation. As shown in the revised Figure 3 and Figure 4, these reporter signals did not undergo obvious reduction due to *iEPF1*. In summary, the results from our large-scale image quantification fully support our original conclusion.

It is very difficult to produce large quantities of highly pure, properly-refolded bioactive EPF1 peptides. The disappearance of MUTE-GFP protein and consequential developmental arrests of meristemoids by induced *EPF1* overexpression and EPF1 peptide treatments are highly reproducible. For this reason, further time-course quantification analyses of Figure 3—figure supplement 2 would add little insight to our findings.

*2) Provide information as to how the marker lines were generated (transformation or crossing), plant generation used, expression levels of the respective tagged proteins and background used (wild type, mutant). It is not always clear what was the genetic background used to express the tagged proteins, for example in Figure 2, are ERL1-YFP and MUTE-tagRFP expressed in the Col-0 or erl1 background? Similarly for Figure 2.*

*ERL1pro::ERL1-YFP* and *MUTEpro::MUTE-tagRFP* are expressed in *erl1* and Col, respectively then subjected to genetic crosses. For all marker lines used for this study, we have provided detailed information about their genetic background in the revised Methods section and figure legend as appropriate.

*3) Fluorescently-tagged receptors are not always functional (sometimes they are only partially functional, for example BAK1, FLS2); therefore is important that the authors quantify the phenotypes of the rescued er erl1 erl2 triple mutant (Figure 5), in order to be more convincing that the ERL1-GFP fusion is functional (similar experiments were done for pMUTE::ERL1-YFP introduced in the er erl1 erl2 triple mutant).*

This is a valid criticism. In the original submission, we provided the confocal image of the cotyledon epidermis showing that *ERL1pro::ERL1-YFP* rescues the stomatal clustering phenotype of *er erl1 erl2* triple mutant (Figure 1—figure supplement 1). As requested, we have performed quantitative analysis of stomatal index and clustering distribution, now provided as Figure 1—figure supplement 1.

*Also, the expression levels should be close to the endogenous, as even when using an endogenous promoter, overexpression of the transgene is frequently observed.*

As requested, we have analyzed the expression level of *ERL1pro::ERL1-YFP* in the *erl1* knockout background (revised Figure 1—figure supplement 1). We detected a slight (~1.25 fold) but not significant elevation of the transgene expression compared to the wild-type *ERL1* (p=0.074, Student's t-test). Since *ERL1pro::ERL1-YFP* rescues the stomatal clustering phenotype of *er erl1 erl2* to the same extent as the endogenous *ERL1* gene (p= 0.49, Tukey's HSD test. See Figure 1—figure supplement 1), the transgene does not confer overexpression or ectopic effects

*Is the MUTE-GFP fusion functional?*

Yes. Complementation of *mute* by *MUTEpro::MUTE-GFP* was published previously (Pillitteri et al., 2007 Nature 445: 501-).

*4) TagRFP is a slowly maturing protein (100 min, in contrast to 10 min for GFP) and this might affect the co-localization dynamics shown in Figure 2, Figure 5. Provide data or discuss this possibility.*

We have performed a careful analysis of *MUTEpro::MUTE-GFP* and *MUTEpro::MUTEtagRFP* expression during stomatal development. As shown in the revised Figure 2—figure supplement 1, their signal accumulation patterns are essentially identical. In addition, during the revision process we did genetic crosses, generated a double transgenic F1 seedlings carrying both *MUTEpro::MUTE-GFP* and *MUTEpro::MUTEtagRFP*, and analyzed their signals. Again, both GFP and RFP signals are co-detected during meristemoid-to-GMC transition (revised Figure 2—figure supplement 1).

Since we are analyzing their accumulation patterns during the merstemoid-to-GMC transition, which takes an average of 19 hours (see Figure 5), we believe that the ~90 minute differences in the maturation time for GFP vs. tagRFP would not affect the co-localization dynamics or interpretation of our results within the time scale of our analysis.

*5) It is not clear why MUTE-GFP in the scrm-D background was used for the ChIP assays and not MUTE-GFP in wild type. Is SCRM required for MUTE binding? Explain or discuss.*

As referenced in our original submission, the effective use of the *scrm-D* background for ChIP experiments was published previously (Horst et al., 2015 PLOS Genetics). The *scrm-D* mutation allows the entire protoderm to undergo stomatal cell-lineage transitions (Kanaoka et al., 2008 Plant Cell). Because nearly all epidermal cells follow the meristemoid-to-GMC transition, one could substantially increase the signals for MUTE-GFP ChIP assays. In the revised manuscript, we have explained rationales behind using the *scrm-D* background (subsection “ERL1 is a direct MUTE target”, second paragraph).

We have in addition performed ChIP experiments using Arabidopsis seedlings carrying functional *MUTEpro::MUTE-GFP* in *mute* null mutant background. This line was published previously, and the seedlings are phenotypically wild type (Pillitteri et al., 2007 Nature). We were able to detect the binding of MUTE-GFP to the *ERL1* promoter region with lesser fold enrichment, owning to fewer cells undergoing meristemoid-to-GMC transition (Revised Figure 2—figure supplement 2). The results further emphasize that MUTE binds to the *ERL1* promoter during normal stomatal development (and therefore it is not the artifact of the *scrm-D* mutation).